# Hidden in the Fat: Unpacking the Metabolic Tango Between Metabolic Dysfunction-Associated Steatotic Liver Disease and Metabolic Syndrome

**DOI:** 10.3390/ijms26073448

**Published:** 2025-04-07

**Authors:** Mariana Boulos, Rabia S. Mousa, Nizar Jeries, Elias Simaan, Klode Alam, Bulus Bulus, Nimer Assy

**Affiliations:** 1Internal Medicine Department, Galilee Medical Centre, Nahariya 221001, Israel; rabia.mousa@gmail.com (R.S.M.); nizar_j_93@hotmail.com (N.J.); eliasol937@gmail.com (E.S.); klod_alam@hotmail.com (K.A.); bulus.91@gmail.com (B.B.); nimera@gmc.gov.il (N.A.); 2The Azrieli Faculty of Medicine, Bar-Ilan University, Safed 1311502, Israel

**Keywords:** metabolic syndrome, MASLD mechanisms, future directions

## Abstract

Metabolic syndrome (MetS) and metabolic dysfunction-associated steatotic liver disease (MASLD) are closely related, with rapidly increasing prevalence globally, driving significant public health concerns. Both conditions share common pathophysiological mechanisms such as insulin resistance (IR), adipose tissue dysfunction, oxidative stress, and gut microbiota dysbiosis, which contribute to their co-occurrence and progression. While the clinical implications of this overlap, including increased cardiovascular, renal, and hepatic risk, are well recognized, current diagnostic and therapeutic approaches remain insufficient due to the clinical and individuals’ heterogeneity and complexity of these diseases. This review aims to provide an in-depth exploration of the molecular mechanisms linking MetS and MASLD, identify critical gaps in our understanding, and highlight existing challenges in early detection and treatment. Despite advancements in biomarkers and therapeutic interventions, the need for a comprehensive, integrated approach remains. The review also discusses emerging therapies targeting specific pathways, the potential of precision medicine, and the growing role of artificial intelligence in enhancing research and clinical management. Future research is urgently needed to combine multi-omics data, precision medicine, and novel biomarkers to better understand the complex interactions between MetS and MASLD. Collaborative, multidisciplinary efforts are essential to develop more effective diagnostic tools and therapies to address these diseases on a global scale.

## 1. Introduction

One of the most important global health issues facing modern medicine is the changing landscape of MASLD and MetS [1,2]. Several studies suggest differences in the prevalence and severity of MASLD by race or ethnicity [3,4,5,6]. According to recent epidemiological data, prevalence varies significantly by geography, with figures ranging from 29.6 percent to 32.2 percent in the Asia–Pacific region, 37.3 percent in North America, and 32.8 percent in Europe, and the highest prevalence of 44.37% was documented among Latin America [7,8]. This distribution pattern illustrates the intricate relationships that exist between lifestyle choices, socioeconomic determinants, and genetic predisposition [9]. According to recent meta-analyses, the global prevalence of the disease is expected to rise by 47% by 2030, with annual incidence rates of 2–8% in Asian populations and 6–9% in Western nations [10,11].

Pro-inflammatory mediators, altered adipokine profiles, and disruption of metabolic homeostasis are all examples of the complex bidirectional pathways that make up the pathophysiological relationship between MetS and MASLD [12,13]. The oxidative stress pathway and IR are two mechanistic paradigms that best illustrate this relationship [14]. Adipose tissue dysfunction sets off a series of events that result in hepatic lipid accumulation and inflammatory activation. It is characterized by dysregulated adipokine secretion and increased lipolysis [15,16]. Furthermore, hepatic steatosis creates a self-reinforcing cycle of metabolic degradation by sustaining systemic metabolic abnormalities via a variety of mechanisms [17,18].

Genetic factors also play an important role in the development of MASLD and can explain the variability in the natural history of MASLD. In studies involving family clusters, one fifth of metabolic dysfunction-associated steatohepatitis (MASH) patients had a similar first-degree relative [19]. Another study reported that seven of the eight families had MASH and cryptogenic cirrhosis [20]. These hypotheses about genetic factors that contribute to MASLD may explain the development and progression of MASLD in lean patients, although the mechanism is still poorly understood in this population. One study reported a higher prevalence of IR in thin Asian men compared to Hispanic, black, and white men [21]. The Genome-Wide-Association Study (GWAS) identified 17 MASLD-associated genes, such as torsin family 1 member B (TOR1B), fat mass and obesity-associated (FTO) gene, growth factor receptor-bound protein 14 (GRB14), insulin receptor (INSR), cordon-bleu WH2 repeat protein like 1 (COBLL1), sterol regulatory element-binding transcription factor 1 (SREBF1), and patatin-like phospholipase domain-containing protein 2 (PNPLA2) [22].

Existing diagnostic approaches often detect these diseases in an advanced stage, which highlights the need for preventive strategies [23]. With new developments in metabolomics, proteomics, and imaging techniques offering promising routes to early detection, the development of new biomarkers and more sensitive diagnostic tools is a major unmet need [24]. Integrating artificial intelligence with traditional diagnostic methods has opened up new possibilities for stratifying risk and diagnosing rare diseases [25].

In view of the rapidly evolving understanding of these interrelated conditions and their significant impact on global health, this review aims to examine in greater detail the molecular mechanisms behind the two-way relationship between METS and MASLD, with a focus on new pathophysiological pathways and clinical implications. It will also explore possible therapeutic targets based on our growing understanding of the common pathophysiology of these conditions. This compilation of the latest data will provide researchers and clinicians with a modern framework for understanding how diseases progress and will guide the development of more effective treatment options.

## 2. Change of Nomenclature

There has been a substantial change in the way the medical community views these conditions with the changing of the term “NAFLD” to “metabolic-dysfunction associated liver disease” (MASLD). In 2020, a global consensus panel suggested changing the name of “NAFLD” to “MASLD” to more accurately represent the disease’s intricate metabolic foundations [26]. By recognizing the importance of IR, obesity, type 2 diabetes, and other metabolic abnormalities in the pathophysiology of fatty liver disease, the new nomenclature highlights the connection between liver disease and metabolic dysfunction [1,27]. The change was prompted by the realization that the term “non-alcoholic” is restrictive because it fails to encompass the wider range of metabolic elements that contribute to the disease. MASLD enables a broader definition that takes into consideration the metabolic environment in which liver steatosis arises and advances.

The reclassification of non-alcoholic steatohepatitis (NASH) to metabolic dysfunction-associated steatohepatitis (MASH) followed, together with the change to MASLD. The switch to MASH highlights the growing understanding that hepatocyte damage and inflammation in steatosis reflect complex interactions between metabolic disorders such as oxidative stress, IR, and adipose tissue dysfunction rather than being the sole consequence of fat accumulation. The purpose of coining the term MASH was to identify metabolic factors that contribute to the progression from simple steatosis to more severe liver diseases such as fibrosis and cirrhosis, and to move away from the restrictive label of alcoholics only. With the official adoption of these changes to the nomenclature in 2020, our understanding of liver disease is changing.

## 3. Underlying Mechanisms

### 3.1. Insulin Resistance: The Central Pathway

IR serves as a cornerstone of both MetS and MASLD, linking these two conditions through multiple interconnected metabolic pathways. In MetS, IR results in a diminished cellular response to insulin, particularly in adipose tissue, muscle, and the liver. This impaired response drives a cascade of metabolic disruptions that promote the development and progression to MASLD [28].

At the hepatic level, IR is a key driver of hepatic steatosis, which is the hallmark of MASLD. Normally, insulin regulates hepatic glucose production and promotes lipid storage through pathways such as de novo lipogenesis. In the insulin-resistant state, these processes become dysregulated, resulting in several pathological changes in the liver. One of the key processes affected is de novo lipogenesis, where chronic hyperinsulinemia stimulates the liver to convert excess carbohydrates into fatty acids. This process is mediated by the upregulation of the sterol regulatory element-binding protein 1c (SREBP-1c) pathway, a transcription factor that promotes fatty acid synthesis [29]. The overproduction of triglycerides, driven by this mechanism, leads to their accumulation in hepatocytes, contributing to hepatic steatosis.

Additionally, IR impairs the normal suppression of lipolysis in adipose tissue. Under normal conditions, insulin inhibits the release of free fatty acids (FFAs) from adipose stores. However, in the context of IR, this suppression is impaired, leading to an increased flux of FFAs into the circulation. These excess FFAs are taken up by the liver, where they further contribute to triglyceride accumulation and exacerbate hepatic steatosis [30]. Furthermore, IR reduces fatty acid β-oxidation in hepatocytes, a process critical for breaking down fatty acids to energy production. This reduction is partly due to the dysregulation of peroxisome proliferator-activated receptor-α (PPAR-α), a nuclear receptor responsible for promoting fatty acid oxidation [31]. The diminished capacity of oxidation leads to the accumulation of lipotoxic intermediates, which further drive hepatocyte injury and inflammation.

In addition to promoting hepatic fat accumulation, IR contributes to a pro-inflammatory state in the liver. This is compounded by the secretion of hepatokines, liver-derived proteins such as fetuin-A, which impair insulin signaling in peripheral tissues and promote systemic inflammation [32]. Fetuin-A, for instance, disrupts insulin receptor signaling in muscle and adipose tissue, creating a feedback loop of worsening IR and metabolic dysfunction.

Moreover, IR is associated with endoplasmic reticulum (ER) stress and oxidative stress, both of which play key roles in the progression to MASLD. The chronic overnutrition and lipid overload, both hallmark features of MetS, compromise the liver’s metabolic capacity, resulting to ER dysfunction and the accumulation of reactive oxygen species (ROS) [33].

These processes contribute to hepatocyte injury and promote progression from simple steatosis to more severe stages of liver disease, such as MASH and cirrhosis. IR is therefore the primary mechanism by which MetS induces MASLD development and progression.

### 3.2. Adipose Tissue Dysfunction and Pro-Inflammatory State

Adipose tissue dysfunction is a hallmark of MetS and plays a critical role in the development of MASLD by promoting both IR and systemic inflammation. In healthy metabolic states, adipose tissue serves as an energy reservoir and a dynamic endocrine organ, releasing a range of adipokines that regulate insulin sensitivity, lipid metabolism, and inflammatory responses. However, in the context of MetS, excess adiposity leads to significant alterations in the secretion of these adipokines, creating a pro-inflammatory and insulin-resistant environment that directly impacts the liver.

One of the key alterations in adipose tissue dysfunction is the imbalance between anti-inflammatory and pro-inflammatory adipokines. In MetS, there is typically a reduction in adiponectin, an adipokine that possesses insulin-sensitizing, anti-inflammatory, and antifibrotic properties. Reduced adiponectin levels are strongly associated with increased hepatic fat accumulation and inflammation, contributing to the development of MASLD. In contrast, pro-inflammatory adipokines such as leptin, tumor necrosis factor-α (TNF-α), and interleukin-6 (IL-6) are elevated in individuals with MetS. These cytokines are known to promote systemic and hepatic IR, as well as hepatic inflammation, both of which are key drivers to MASLD onset and progression [34,35].

Leptin, although primarily involved in regulating appetite and energy homeostasis, becomes dysregulated in the context of obesity and MetS. Elevated leptin levels, coupled with the development of leptin resistance, impair the normal metabolic functions of this hormone. Leptin resistance contributes to ectopic lipid accumulation in organs such as the liver, thereby exacerbating hepatic steatosis and inflammation. This leptin dysfunction further feeds into the systemic inflammatory milieu that characterizes MetS, contributing to the worsening of IR [36].

Adipose tissue dysfunction also plays a role in the infiltration of immune cells, particularly macrophages, into adipose depots. These immune cells contribute to the secretion of pro-inflammatory cytokines that exacerbate systemic inflammation and liver injury. In obesity and MetS, macrophages undergo a phenotypic switch to a more pro-inflammatory, M1-like state, characterized by the production of elevated levels of TNF-α and IL-6. These cytokines not only impair insulin signaling but also promote the activation of inflammatory pathways in the liver, leading to hepatocyte injury, steatosis, and the progression to MASH [32,33].

Moreover, excess adipose tissue, particularly visceral adiposity, contributes to increased FFA release into the circulation due to impaired suppression of lipolysis by insulin. The elevated levels of circulating FFAs are taken up by the liver, where they serve as substrates for triglyceride synthesis, leading to hepatic steatosis. This excess FFA influx also exacerbates hepatic IR, further driving fat accumulation and inflammation in the liver. The combination of these factors—altered adipokine secretion, immune cell infiltration, and increased FFA release—creates a chronic pro-inflammatory state that fosters the progression of MASLD [31].

In summary, the dysfunction of adipose tissue in MetS plays a pivotal role in the pathogenesis of MASLD by promoting systemic inflammation, IR, and driving hepatic fat accumulation. This pro-inflammatory environment not only contributes to hepatic steatosis but also facilitates the progression of liver disease toward more severe forms, including MASH and fibrosis.

### 3.3. ER Stress and Oxidative Stress: Amplifying Liver Injury

In the context of MetS and MASLD, ER stress and oxidative stress play key roles in amplifying liver injury and promoting the progression from simple steatosis to MASH. Both processes are tightly linked to the metabolic dysregulation seen in MetS, contributing to hepatocyte dysfunction, inflammation, and fibrosis.

When ER’s folding capacity is overwhelmed due to excessive nutrient load, lipid accumulation, ER stress occurs which leads to ER lumen accumulation of misfolded or unfolded proteins, triggering the unfolded protein response (UPR). While the latter is protective initially with the aim to restore normal ER function, prolonged or unresolved ER stress can lead to apoptosis and inflammation. In MetS, lipid overload and chronic overnutrition overwhelm the liver’s ability to maintain protein homeostasis leading to subsequent hepatocyte injury [32].

In parallel with ER stress, oxidative stress plays a significant role in the pathogenesis of MASLD. Oxidative stress occurs when the production of ROS exceeds the liver’s antioxidant defenses, leading to cellular damage. In MetS, the increased flux of free FFAs to the liver, coupled with impaired fatty acid oxidation, results in excessive β-oxidation in mitochondria. This overloads the mitochondrial electron transport chain, leading to the excessive production of ROS, particularly superoxide anions [37].

These ROS not only directly damage cellular components, including lipids, proteins, and Deoxyribonucleic acid (DNA), but also activate inflammatory pathways. Oxidative stress has been shown to activate the NACHT, LRR, and PYD domain-containing protein 3 (NLRP3) inflammasome, a multiprotein complex that promotes the production of pro-inflammatory cytokines such as interleukin-1β (IL-1β) and interleukin 18 (IL-18) [33]. The activation of the NLRP3 inflammasome contributes to the progression of simple steatosis to MASH, as it fosters a pro-inflammatory environment that perpetuates hepatocyte injury and fibrosis.

Moreover, oxidative stress and ER stress are mutually reinforcing processes. The accumulation of misfolded proteins in the ER can trigger oxidative stress by disturbing calcium homeostasis and increasing ROS production. Conversely, oxidative damage to proteins and lipids can further impair ER function, exacerbating the unfolded protein response and driving further hepatocyte injury [38]. This interplay between ER stress and oxidative stress represents a critical mechanism underlying the progression of MASLD, as it leads to both metabolic dysfunction and chronic liver inflammation.

In summary, both ER stress and oxidative stress are key amplifiers of liver injury in MetS and MASLD. These processes not only exacerbate IR and lipid accumulation but also drive inflammation and hepatocyte apoptosis, contributing to the progression from simple steatosis to MASH and fibrosis. Addressing these stress responses offers potential therapeutic avenues for mitigating the liver damage associated with MASLD.

### 3.4. Gut Microbiota Dysbiosis and Metabolic-Endotoxemia

The gut microbiota has emerged as a key player in the pathogenesis of both MetS and MASLD. Dysbiosis, or the imbalance in gut microbial composition, is linked to metabolic disturbances that characterize MetS, including IR, obesity, and chronic inflammation. In recent years, evidence has increasingly supported the notion that alterations in the gut microbiota contribute to the development of MASLD by promoting hepatic steatosis, inflammation, and fibrosis.

One of the primary mechanisms by which gut dysbiosis drives liver pathology is through increased intestinal permeability, often referred to as “leaky gut”. In healthy individuals, the intestinal barrier limits the translocation of bacterial products into the circulation. However, in individuals with MetS, alterations in the gut microbiota compromise the integrity of the intestinal barrier, allowing bacterial endotoxins such as lipopolysaccharides (LPS) to enter the portal circulation. This process, known as metabolic endotoxemia, plays a central role in the development of MASLD [39].

LPS, a component of the outer membrane of Gram-negative bacteria, is a potent activator of the innate immune system. Once translocated into the liver, LPS binds to toll-like receptor 4 (TLR4) on Kupffer cells (KCs) and hepatocytes, triggering the production of pro-inflammatory cytokines such as TNF-α and IL-6. This inflammatory response promotes hepatic IR and contributes to hepatocyte injury, steatosis, and fibrosis [40]. Additionally, metabolic endotoxemia perpetuates a cycle of chronic, low-grade inflammation that exacerbates the metabolic disturbances associated with MetS.

Beyond LPS, gut dysbiosis also impacts the metabolism of BAs, which are crucial regulators of lipid and glucose homeostasis. The gut microbiota plays a key role in transformation and deconjugation of BAs, by affecting the signaling through receptors such as the Farnesoid X receptor (FXR) and G protein-coupled bile acid receptor 1 (TGR5). Dysbiosis causing alterations in the BA pool might lead to impaired lipid metabolism and increased hepatic fat accumulation. In the setting of MASLD, disrupted BA signaling contributes to IR, hepatic steatosis, and inflammation [41].

Moreover, the gut microbiota produces a variety of microbial metabolites that can influence host metabolism. Short-chain fatty acids (SCFAs), which are produced by the fermentation of dietary fiber, have been shown to enhance insulin sensitivity and exert anti-inflammatory effects. However, in the context of dysbiosis, the production of SCFAs may be reduced, contributing to metabolic dysfunction and inflammation. Conversely, other microbial metabolites, such as imidazole propionate, have been shown to impair insulin signaling and promote IR, further linking gut dysbiosis to the pathogenesis of MetS and MASLD [32,42].

In summary, gut microbiota dysbiosis plays a vital role in the development and progression of MASLD through mechanisms involving metabolic endotoxemia, BA dysregulation, and altered microbial metabolite production. These factors contribute to hepatic steatosis, IR, and inflammation, highlighting the gut–liver axis as a potential therapeutic target in the management of MASLD.

### 3.5. Innate Immune Response and Hepatic Inflammation

The innate immune response plays a significant role in the pathogenesis of MASLD, particularly in driving hepatic inflammation and promoting the progression from simple steatosis to MASH. In individuals with MetS, the liver becomes a key site of immune cell activation, leading to a chronic inflammatory state that exacerbates IR, hepatocyte injury, and fibrosis.

At the forefront of the innate immune response in the liver are Kupffer cells, the liver’s resident macrophages. In a healthy liver, KCs maintain tissue homeostasis by clearing debris and apoptotic cells while preventing excessive immune activation. However, in the context of MetS and MASLD, KCs become activated in response to metabolic stressors, such as FFAs and microbial products translocated from the gut. This activation leads to the production of pro-inflammatory cytokines, such as TNF-α, IL-1β, and IL-6, that promote hepatic inflammation and IR [33,42].

The activation of KCs is also influenced by LPS, bacterial endotoxins that enter the liver via the portal circulation during a metabolic endotoxemia state. LPS binds to TLR4 on KCs and other hepatic immune cells, triggering downstream signaling pathways that enhance the production of inflammatory cytokines. This LPS-mediated activation of TLR4 contributes to the amplification of hepatic inflammation in MASLD and fosters a microenvironment conducive for the progression of steatosis to MASH [39].

In addition to Kupffer cells, infiltrating monocytes and macrophages play a crucial role in the innate immune response in the liver. In response to signals from damaged hepatocytes and adipose tissue, monocytes are recruited to the liver, where they differentiate into pro-inflammatory macrophages. These macrophages, often referred to as M1-like macrophages, produce elevated levels of pro-inflammatory cytokines and contribute to the ongoing cycle of inflammation and liver damage. The accumulation of M1-like macrophages in the liver is a hallmark of MASH and is closely linked to the progression of liver fibrosis [32,40].

In summary, the innate immune response, driven by Kupffer cells, macrophages, and the NLRP3 inflammasome, plays a pivotal role in the pathogenesis of MASLD. Through the production of pro-inflammatory cytokines and the activation of fibrotic pathways, innate immune cells contribute to the progression of liver disease from simple steatosis to MASH and fibrosis. Targeting these immune pathways represents a potential therapeutic strategy for halting the progression of MASLD.

### 3.6. Fibrosis Pathways and Liver Disease Progression

Fibrosis is a hallmark of the progression of MASLD from simple steatosis to more advanced stages of liver disease, MASH, advanced fibrosis, and cirrhosis. The development to fibrosis is driven by chronic inflammation and metabolic dysregulation, both of which are characteristic features of MetS. In MASLD, fibrosis occurs because of the excessive deposition of extracellular matrix (ECM) components, including collagen, which leads to scarring and architectural distortion of the liver. This process is orchestrated by hepatic stellate cells (HSCs), which become activated in response to injury signals from damaged hepatocytes, immune cells, and the metabolic environment.

In healthy liver tissue, HSCs remain in a quiescent state and function primarily in vitamin A storage. However, in the setting of MASLD, persistent metabolic stress, lipid accumulation, and chronic inflammation lead to the activation of HSCs. Once activated, HSCs transform into myofibroblast-like cells that secrete substantial amounts of ECM proteins, including collagen types I and III, which contribute to the development of fibrotic tissue. This activation process is mediated by a variety of signaling molecules, including transforming growth factor-β (TGF-β), which is the most potent profibrogenic cytokine in the liver [33]. TGF-β is secreted by Kupffer cells, activated HSCs, and infiltrating immune cells in response to hepatic injury. Through the TGF-β/Smad signaling pathway, HSCs are stimulated to proliferate and synthesize ECM components, leading to progressive fibrosis.

In addition to TGF-β, other growth factors and cytokines contribute to the activation of HSCs and the fibrotic process. Connective tissue growth factor (CTGF), also known as Cellular communication network 2 (CCN2), plays a significant role in amplifying the fibrogenic response in the liver. CTGF is upregulated in response to TGF-β and other pro-fibrotic stimuli and enhances the production of collagen and other ECM proteins by HSCs [38]. Furthermore, platelet-derived growth factor (PDGF), which is released by platelets, Kupffer cells, and damaged hepatocytes, acts as a potent mitogen for HSCs, stimulating their proliferation and migration within the liver. Together, these signaling pathways create a pro-fibrogenic environment that promotes the accumulation of scar tissue and the progression of MASLD to cirrhosis.

The progression of fibrosis in MASLD is a major determinant of patient outcomes, as advanced fibrosis is associated with an increased risk of liver-related complications, including hepatocellular carcinoma (HCC), portal hypertension, and liver failure [43]. Studies have shown that patients with advanced fibrosis, particularly those with bridging fibrosis or cirrhosis, present significantly higher mortality rates compared to those with mild or no fibrosis [44]. As such, targeting the fibrotic pathways involved in HSC activation and ECM deposition represents a key therapeutic strategy for preventing the progression of MASLD.

To summarize, fibrosis in MASLD is driven by a complex interplay of metabolic dysregulation, chronic inflammation, and the activation of pro-fibrogenic signaling pathways. TGF-β and CTGF are central mediators of HSC activation and ECM deposition, while chronic inflammation perpetuates the fibrotic response. Understanding these mechanisms is crucial for the development of effective therapies aimed at halting or reversing fibrosis in MASLD.

In summary, MASLD and MetS are tightly linked by IR, which disrupts hepatic glucose/lipid metabolism, leading to fat accumulation and a pro-inflammatory state amplified by adipose tissue dysfunction, ER/oxidative stress, and gut dysbiosis. These factors activate the innate immune response in the liver, resulting in fibrosis, the key process in MASLD progression to severe liver disease (Figure 1).

## 4. Extrahepatic Co-Morbidities

Beyond hepatic symptoms, MetS and MASLD represent a substantial health burden with extensive systemic ramifications. The combinations of these disorders form a complex and vicious environment that impacts the cardiovascular, renal, endocrine, and immune systems, among other organ systems. Comprehending these systemic effects is essential for both optimal clinical outcomes and comprehensive patient care [45,46].

### 4.1. Cardiovascular Morbidity and Mortality

Cardiovascular disease (CVD) is still the world’s leading cause of death, with metabolic syndrome being strongly associated with an increased risk. Cardiovascular outcomes are doubled in patients with MetS, and all-cause mortality is increased by one to five times [47].

This risk is increased further when MASLD is present. The Targher et al. meta-analysis reported that regardless of conventional cardiovascular risk factors or the elements of MetS, MASLD was linked to a 64% higher risk of fatal and/or non-fatal CVD events (2016) [47,48]. Even after controlling for changes in Body Mass Index (BMI), progression of cardiovascular risk factors was linked to rising hepatic fat over a 6-year period in a longitudinal cohort [49]. Additionally, in a prospective research study, Basheer M et al. concluded that a strong independent risk factor for coronary artery disease (CAD) and carotid atherosclerosis is liver fat storage more than visceral fat storage [50]. Nevertheless, epicardial adipose tissue (EAT) was found to be significantly higher in MASLD patients than in controls in a meta-analysis involving 13 case–control studies (*n* = 2260 patients). Moreover, EAT was associated with the degree of atherosclerotic CVD and liver steatosis and fibrosis [51]. Nonetheless, it is the visceral fat depot of the heart that produces both pro-inflammatory and anti-inflammatory mediators [52]. According to one study involving 868 participants, an increased incidence of extra-cardiac plaques was associated with hepatic steatosis and epicardial fat thickness [53].

MASLD increases the risk of CVD through a number of mechanisms, including systemic low-grade inflammation, adhesion molecules, hepatic IR, and a prothrombotic state [54]. A notable reduction in brachial artery endothelial flow-mediated vasodilatation is another indication of endothelial dysfunction in people with MASLD [55]. Additionally, through the liver’s secreted proteins, including IL-6, CRP, fibrinogen, Monocyte chemoattractant protein-1 (MCP-1), TNF-α, β-trophin, and fetuin-A, MASH may contribute to systemic low-grade inflammation and cardio-metabolic disease [56].

In conclusion, metabolic disorders like obesity, dyslipidemia, hypertension, and diabetes are strongly associated with MASLD and MASH. An increased risk of CVD is linked to the existence and severity of MASLD, specifically MASH and fibrosis. However, the coexistence of type 2 diabetes (T2D) or a metabolic syndrome is necessary for this association to exist. Significantly, MASLD raises the risk of arrhythmia, cardiomyopathy, and atherosclerosis on its own, which may lead to cardiovascular morbidity and death [57]. Given the strong correlation between obesity and cardio-metabolic disorders, diet and exercise-based lifestyle interventions to promote weight loss are currently the gold standard for treating these conditions.

### 4.2. Progression of Liver Disease

The development of hepatic steatosis into MASH is strongly associated with the level of metabolic dysfunction. Hepatocellular ballooning, inflammatory infiltration, and progressive fibrosis are some of the changes that have occurred [44]. Clinical evidence indicates that patients with metabolic syndrome exhibit an accelerated disease trajectory, with type 2 diabetes mellitus associated with a 1.5–2.0-fold increase in fibrosis progression rate compared to non-diabetic individuals [58]. The combined impact of several metabolic risk factors also shows synergistic potential in the progression of the disease, increasing the risk of liver-related mortality, cirrhosis, and hepatocellular carcinoma [59].

Multiple pathogenic mechanisms beyond simple fat accumulation are involved in the progression of liver disease in MASLD [60]. In hepatocytes, lipotoxicity, which is caused by the buildup of toxic lipid species, sets off cellular stress reactions and death pathways [61]. Chronic inflammation and tissue damage result from this process activation of inflammatory cascades and encouragement of immune cell recruitment [62]. In addition to causing significant alterations in the intestinal barrier function and the gut microbiota’s composition, MASLD also raises endotoxemia and bacterial translocation, both of which exacerbate hepatic inflammation and fibrosis [63].

The progression of MASLD to advanced liver disease is characterized by molecular signatures and distinct histopathological changes [64]. Hepatocyte ballooning, progressive fibrosis, and inflammatory cell infiltration become noticeable characteristics as the disease progresses, although simple steatosis is present in the initial stages. The progression from MASLD to MASH and ultimately HCC is influenced by a number of factors, such as oxidative stress, mitochondrial dysfunction, cellular senescence, and genetic predisposition (PNPLA3 and Transmembrane 6 superfamily member 2 (TM6SF2) variants) [65]. Hepatocyte death and the recruitment of inflammatory cells are also influenced by altered hepatic insulin signaling and chronic endoplasmic reticulum stress [60]. The accumulation of genetic changes in an inflammatory milieu, along with the aberrant activation of oncogenic pathways, specifically Phosphatidylinositol-4,5-bisphosphate 3-kinase (PI3K)/protein kinase B (Akt) PI3K/AKT and mammalian target of rapamycin mTOR signaling, further promotes the progression to HCC [66]. Influenced by both environmental and genetic factors, the rate of progression varies significantly among individuals [67]. About 20% of MASLD patients, according to recent data, progress to MASH, and of these, up to 20% may develop advanced fibrosis in a comparatively short period of time—two to three years—with an annual incidence of 2–3 percent for HCC in patients with cirrhosis related to MASH [9,68].

In summary, a harmful cycle of increasing severity is established by the reciprocal relationship between metabolic syndrome and the progression of hepatic disease. Hepatic injury is accelerated by systemic IR and metabolic disturbances that are made worse by progressive liver dysfunction.

### 4.3. Type 2 Diabetes Mellitus

New definitions were developed through international consensus to better characterize the complex interrelationships between MetS, MASLD, and their combined impact on T2D. This presents a significant challenge in modern medicine [45]. These conditions have been shown to have a synergistic effect that significantly affects glycemic control, the course of the disease, and the results of treatment for diabetic patients in meta-analyses involving over 500,000 adults [69].

Extensive clinical studies have shown that the primary manifestation of the impact on glycemic control is impaired hepatic glucose regulation [70]. Recent studies have demonstrated that the liver’s reduced response to insulin causes increased hepatic glucose output despite hyperglycemia, impaired glycogen storage, and altered glucose homeostasis. MASLD significantly impairs hepatic insulin sensitivity, resulting in dysregulated glucose production and altered metabolic pathways [71]. MetS compounds these effects through adipose tissue dysfunction, increased inflammatory cytokine production, and altered adipokine profiles, as documented in systematic reviews [72]. Clinical research has shown that a more severe form of IR that impacts several tissues and metabolic pathways at once results from the coexistence of both conditions [73].

The disease progression mechanisms show that the combination of MASLD and MetS accelerates beta-cell dysfunction through several pathways [74,75]. Beta-cell function deterioration is caused by oxidative stress, endoplasmic reticulum stress, chronic inflammation, and lipotoxicity from elevated free fatty acids, according to current research [76]. This creates a vicious cycle where declining beta-cell function necessitates increasingly aggressive therapy [74].

Vascular complications are a major concern as well. Recent research indicates that both conditions activate the polyol pathway, produce advanced glycation end products, generate reactive oxygen species and oxidative stress, activate the hexosamine pathway, cause endothelial dysfunction, and cause IR, among other mechanisms that accelerate vascular damage. Further studies have shown that these illnesses can contribute to diabetic patients’ chronic kidney disease on their own [77]. The clinical outcomes present both short-term and long-term challenges, with systematic reviews demonstrating that patients experience greater difficulty achieving glycemic targets and more frequent complications [78]. Reduced quality of life, elevated cardiovascular risk, higher mortality rates, and accelerated progression of complications have all been documented in long-term studies [79].

### 4.4. Extrahepatic Complications

Beyond its effects on cardiovascular health and diabetes risk, MASLD is associated with a range of extrahepatic complications, many of which are also components or consequences of MetS, some of which are described below.

#### 4.4.1. Chronic Kidney Disease (CKD)

The elevated incidence and prevalence of CKD are independently linked to MASLD. Patients with more severe forms of MASLD are especially at risk of CKD [80]. Chen et al. published a fascinating study in *Hepatology International* that included 337,783 participants from the United Kingdom (UK) Biobank who were tracked for a median of 8–12 years. The study found that MASLD with both CKD and non-CKD participants had a twofold increased risk of end-stage kidney disease (ESKD) compared to those without the condition. As liver fibrosis scores increase, the association increases. The polygenic risk alleles PNPLA3 rs738409, TM6SF2 rs58542926, Glucokinase regulatory protein (GCKR) rs1260326, and Membrane-bound O-acyltransferase 7 (MBOAT7) rs641738 were also shown to enhance the MASLD effect on ESKD [81].

It is not entirely clear how MASLD and ESKD are related. It is reasonable to assume, nevertheless, that CKD and MASLD are two sides of the same coin. The etiology of both MASLD and CKD is typically similar, and the majority of those affected have metabolic comorbidities like atherogenic dyslipidemia, T2D, obesity, or hypertension [82]. These disorders have remarkably similar molecular mediators, underlying metabolic pathways, and molecular pathophysiologic mechanisms. These include ectopic fat deposition, IR, inflammation, macrophage activation, oxidative stress, and gut dysbiosis [83].

#### 4.4.2. Obstructive Sleep Apnea (OSA)

There is a strong association between MASLD and OSA, with each condition exacerbating the other. OSA-suffering MASLD patients have more severe liver disease. Intermediary mechanisms between OSA and MASLD seem to be similar. OSA is associated with obesity, dyslipidemia, and metabolic syndrome, a well-known comorbidity of MASLD. In addition, OSA results in IR, an oxidative stress state, metabolic lipid disturbances, and gut barrier dysfunction [84]. Therefore, it has been proposed that the pathophysiology and aggravation of the severity of MASLD are related to the presence of OSA, but particularly to its consequence, severe chronic intermittent hypoxia (CIH) [85]. Most research shows a connection between these two long-term illnesses. Türkayet al. demonstrated that, independent of BMI, the presence of OSA increased the severity of steatosis as assessed by ultrasound [86]. According to a recent meta-analysis by Sookoian and Piorola, there were 404 control subjects and 668 OSA patients in 11 studies, consistently using a non-invasive surrogate of MASLD; they showed that OSA was linked to a significant increase in liver enzymes, specifically alanine transaminase (ALT), indicating that OSA is linked to liver deterioration regardless of BMI and diabetic status [87]. According to other recent research, having OSA, with or without CIH which is indicative of OSA severity, increased the severity of MASLD and promoted the development of MASH [88,89,90]. Morbidly obese patients with liver fibrosis had more severe oxygen desaturations and a higher apnea–hypopnea index (AHI) than those without the condition according to a recent study [88]. An interesting finding is that MASLD was identified using a non-invasive systemic algorithm in an OSA population with a broad range of BMIs [from lean to morbidly obese] (i.e., E. With the severity of OSA, the prevalence of MASLD rose (Fibrotest^®^, MASHtest^®^, and Steatotest^®^) [91]. Crucially, steatosis and the degree of nocturnal hypoxia were linked independently [92,93]. The pathophysiology of this interaction is multifactorial, with CIH inducing metabolic abnormalities such as uncontrolled glucose level by increasing the IR [94]. CIH induces gene expression involved in lipogenesis [95], CIH induces Lysyl oxidase (LOX) expression which is involved in extra-cellular matrix rigidity [96], it also induces oxidative stress and lipid peroxidation [97], mitochondrial dysfunction [98], and finally, CIH induces intestinal permeability and disrupts the gut–liver axis [99], which were all discussed earlier as causative mechanisms of MASLD.

#### 4.4.3. Osteoporosis

The relationship between MASLD and osteoporosis is complex and multifactorial. Several recent studies support the association between a decreased bone mineral density and MASLD [100,101,102].

Several mechanisms are suggested to contribute to this association. One potential mechanism is the disruption of vitamin D metabolism. Vitamin D plays a crucial role in calcium absorption and bone health. Studies have shown that individuals with MASLD often have vitamin D deficiency, which can lead to decreased bone mineral density and increased risk of fractures [103,104]. Another potential mechanism is the impact of MASLD on IR and glucose metabolism. IR is a common feature of MASLD and is associated with increased bone resorption and decreased bone formation [105,106]. Additionally, elevated levels of inflammatory markers, often seen in MASLD, can also contribute to bone loss.

Furthermore, the presence of advanced liver disease, such as cirrhosis, may further increase the risk of osteoporosis [107]. Cirrhosis can lead to hormonal imbalances, such as estrogen deficiency in women, which can significantly impact bone health [108].

In summary, mounting data points to a close connection between MASLD and osteoporosis development. The underlying mechanisms are intricate and could include hormone imbalances, IR, inflammation, and vitamin D deficiency. It is important for people with MASLD to be aware of this potential risk and discuss with their doctors’ ways to maintain healthy bones, such as getting enough exercise, consuming adequate calcium and vitamin D, and taking the appropriate medication when necessary.

#### 4.4.4. Endocrine Disorders

Different studies have suggested that MASLD is associated with different endocrine axes. Accumulating clinical and experimental studies have reported that MASLD is associated with polycystic ovarian syndrome (PCOS) and other endocrine disorders. These associations are primarily mediated through shared pathophysiological mechanisms such as IR, hormonal imbalances, and metabolic dysregulation.

##### Polycystic Ovary Syndrome

This syndrome is characterized by ovulatory dysfunction, hyperandrogenism, and polycystic ovarian morphology [109]. The relationship between MASLD and PCOS is bidirectional, with IR serving as a central link. In PCOS, ovarian androgen production is enhanced by hyperinsulinemia, which results in hyperandrogenism. Hepatic steatosis is worsened by this hormonal imbalance by promoting lipid accumulation in the liver. The other way around, MASLD exacerbates IR as discussed above, further aggravating PCOS symptoms [110]. A randomization analysis reported genetic evidence of a causal association between MASLD and PCOS, emphasizing the closely-connected nature of these conditions [111].

##### Thyroid Dysfunction

The thyroid axis has extreme effects on energy metabolism, fatty acid oxidation, and hepatic lipogenesis. The liver not only receives signals from thyroid hormones but also has receptors for thyroid-stimulating hormone (TSH), an agonist which induces hepatic steatosis. In addition, thyroid hormones are also crucial in regulating metabolism, and their imbalance can significantly impact liver function. Hypothyroidism, characterized by reduced thyroid hormone levels, results in a decreased basal metabolic rate, lipid accumulation, and weight gain [112], therefore increasing the risk of MASLD. Conversely, MASLD-associated metabolic disturbances and an inflammatory state can impair thyroid function, suggesting a bidirectional relationship. Recent studies have explored the potential of thyroid hormone receptor-beta agonists as therapeutic agents for MASLD, aiming to harness the metabolic regulatory functions of thyroid hormones to ameliorate liver steatosis and inflammation. Thyroid hormone receptor (THR)-β is responsible for regulating metabolic pathways in the liver and is frequently impaired in MASH. On 14 March 2024, the United States Food and Drug Administration (FDA) approved the first treatment for MASH [106]; MAESTRO-NASH reported that resmetirom was superior to a placebo in MASH resolution and improvement in liver fibrosis by at least one stage [107].

In summary, MASLD’s interplay with endocrine disorders such as PCOS and thyroid dysfunction highlights the necessity for a multidisciplinary approach in managing affected individuals. Addressing these interconnected conditions holistically can lead to improved metabolic health and better overall patient outcomes.

#### 4.4.5. Implications for Clinical Management

The diverse clinical effects of METS and MASLD underscore the need for a comprehensive, multidisciplinary approach to patient management. Given the potential for disease progression and development of serious complications, early detection and intervention are critical [113]. Regular screening for METs components in MASLD patients and vice versa is essential. In addition, given the increased cardiovascular risk, aggressive management of cardiovascular risk factors in all patients with MetS and MASLD is warranted. These include lifestyle changes such as weight loss and increased physical activity, as well as pharmacological interventions where appropriate [114]. The complex interplay of these disorders also underscores the importance of a personalized medical approach. Factors such as genetic predisposition, environmental influences, and the presence of comorbidities should be considered when developing treatment strategies for individual patients [115].

In conclusion, the clinical implications of MetS and MASLD are far-reaching, affecting multiple organ systems and significantly impacting patient morbidity, mortality, and quality of life. A thorough understanding of these implications is crucial for healthcare providers to deliver optimal care and improve patient outcomes [116].

## 5. Therapeutic Strategies and Future Directions

The treatment of MetS and MASLD requires a multi-layered approach that targets various aspects of these complex disorders. Recent advances in our understanding of its pathophysiology have led to the development of novel therapeutic strategies and potential future treatments. In this section, the mechanisms of action of the individual treatment strategies and their specific significance for both MetS and MASLD are explained in more detail.

### 5.1. Lifestyle Modifications

Lifestyle interventions remain the cornerstone of treatment for MetS and MASLD. Weight loss through calorie reduction and increased physical activity has been shown to improve all components of METs and reduce liver steatosis [117].

Weight loss improves insulin sensitivity, reduces adipose tissue inflammation, and decreases the flow of free fatty acids to the liver. In the context of MetS, this results in improvements in blood pressure, lipid profiles, and glucose metabolism. In MASLD, weight loss reduces liver fat levels, improves liver enzyme levels and, in many cases, can lead to the abatement of MASH [118]. Recent studies have focused on the type of diet that may be most beneficial. The Mediterranean diet, characterized by a high consumption of olive oil, nuts, vegetables, fruits, and fish, has shown promise for improving METs and MASLD. The Mediterranean diet is rich in mono-unsaturated fatty acids, omega-3 polyunsaturated FA, and antioxidants. These components have an anti-inflammatory effect, improve insulin sensitivity, and promote a favorable composition of the intestinal microbiome. In MetS, this diet improves lipid profiles and insulin sensitivity. In MASLD, it reduces liver fat content and improves liver enzyme levels [119,120,121].

Intermittent fasting has also gained attention. Some studies suggest benefits for insulin sensitivity and liver steatosis. Intermittent fasting can improve insulin sensitivity by promoting metabolic changes between glucose and fatty acid oxidation [122]. It may also activate cellular stress response pathways that improve overall metabolic health. For MASLD, intermittent fasting may reduce liver fat levels by promoting fat oxidation during fasting [123].

Physical activity has a variety of effects on MASLD. It increases insulin sensitivity and enables tissue to absorb glucose more efficiently. This reduces glucose production in the liver and decreases the substrate for de novo lipogenesis, an important metabolic pathway in fat accumulation in the liver [124]. Regular physical activity stimulates the breakdown of FAs for energy consumption, a process known as fatty acid oxidation that directly reduces liver fat levels [125]. Exercise also has powerful anti-inflammatory effects and alleviates systemic and liver inflammation, which play an important role in MASLD pathogenesis [126]. In addition, physical activity promotes weight loss or weight maintenance, which is critical for improving MASLD. Even modest weight loss can significantly reduce liver fat levels and improve liver function [127].

Both aerobic and resistance training have demonstrated beneficial effects on MASLD. Aerobic exercise, such as brisk walking, jogging, swimming, and cycling, enhances cardiovascular fitness, improves insulin sensitivity, and promotes fat oxidation [128]. Resistance training, including weightlifting and bodyweight exercises, increases muscle mass, boosts metabolism, and improves insulin sensitivity [129]. A combination of both aerobic and resistance training is often recommended for optimal outcomes [130].

### 5.2. Pharmacological Interventions

#### 5.2.1. Insulin Sensitizers

Metformin, a first-line treatment for T2D, increases peripheral (including hepatic) insulin sensitivity, reduces hepatic glycogenesis, and increases glucose uptake and utilization of insulin-stimulated peripheral tissues in MASLD patients [131]. A meta-analysis suggests that low-dose metformin (500–3000 mg/d) over a short-term period (up to 6 months) can reduce liver enzymes (ALT, aspartate aminotransferase (AST)), triglycerides (TG), and total cholesterol (TC), while improving IR (HOMA-IR) [132]. The improvement in liver function and lipid metabolism appears to be independent of its glucose-lowering effects. Both ALT and AST levels decrease with higher metformin doses and prolonged use, indicating reduced liver damage.

Metformin is also linked to a reduction in TG levels in patients with MASLD [133]. The liver’s ability to balance fat storage and breakdown can be disrupted in metabolic disorders, leading to fat accumulation in the liver and the development of fatty liver. MASLD is associated with dyslipidemia, high blood sugar, and IR. Elevated TG levels can worsen IR and promote a cycle of increasing liver fat [134]. TC, which includes both esterified cholesterol and free (unesterified) cholesterol “cholesteryl ester”, reflects lipid metabolism and can indicate liver damage when elevated in MASLD patients [135]. It has been reported that metformin significantly improves blood lipid levels, including TC and TG, in MASLD patients. Moreover, metformin helps reduce IR and improve liver function, highlighting its potential role in treating MASLD and its associated metabolic issues [136].

The difference between preclinical and clinical outcomes could be due to different disease severity in human patients or differences in metformin absorption between species. Further research, particularly with larger sample sizes, is needed to clarify these effects.

While metformin improves liver parameters in diabetic MASLD patients, newer antidiabetic drugs have shown stronger effects: thiazolidinediones (TZD), Glucagon-like peptide-1 (GLP-1) receptor agonists, and Sodium-Glucose Linked Transporter (SGLT2) inhibitors have shown greater effectiveness in reducing liver fat and improving liver biochemistry than metformin alone; dipeptidyl peptidase 4 (DPP-4) inhibitors (such as sitagliptin) were less effective at reducing liver enzymes as metformin [132] and Sulfonylureas (e.g., gliclazide) also reduced liver fat but to a lesser extent than metformin [137]. In addition, the combination of metformin with other antidiabetic agents has shown additional advantages over monotherapies: TZDs, GLP-1 receptor agonists, DPP-4 inhibitors, and SGLT2 inhibitors, when used together with metformin, improved liver parameters more effectively than metformin alone [138,139].

Thiazolidinediones, nuclear receptors, known as PPARs, are found in adipose tissue, liver, muscle, heart, and kidneys, among others, where genes associated with fatty acid oxidation, lipid transport, and gluconeogenesis are regulated. α, β/β, and ω are the three PPAR subtypes [140,141,142]. Some PPAR agonists are used to treat T2DM, such as pioglitazone and rosiglitazone [143,144].

Pioglitazone, an agonist for PPAR-α, modulates key genes in lipid metabolism [145]. Through increased expression of Apo AI and AII, it lowers triglyceride levels in lipoproteins, increases fatty acid oxidation, promotes fatty acid transporters, decreases ketogenesis, and increases high-density lipoprotein (HDL-C) synthesis [146]. In the pioglitazone versus Vitamin E versus Placebo for the Treatment of Non-Diabetic Patients with Nonalcoholic Steatohepatitis (PIVENS) trial, despite the fact that pioglitazone did not have any benefit in the primary outcome over the placebo, it was associated with a highly significant inflammation reduction, hepatocellular ballooning, and reduction in steatosis, and lower IR and liver enzyme levels [147]. Various studies, including Belfort et al. and Aithal, showed that pioglitazone significantly improved liver histology in MASH patients, particularly obese patients with glucose intolerance [147,148].

As for rosiglitazone, studies have shown that it can reduce hepatic fat accumulation by enhancing adipocyte differentiation and lipid storage in peripheral fat depots, thereby reducing ectopic lipid deposition in the liver. Additionally, rosiglitazone has anti-inflammatory properties, which may help mitigate liver inflammation and fibrosis associated with MASH [149,150]. Several clinical trials have assessed rosiglitazone in MASLD/MASH, with some demonstrating histological improvements in steatosis and inflammation. The Fatty Liver Improvement with Rosiglitazone Therapy (FLIRT) trial reported improvement in transaminase levels and steatosis but also weight gain with no improvement of other liver injury parameters [151]. The continuous FLIRT-2 study reported a substantial steatosis effect within one year though maintaining transaminase levels and an IR effect [152]. Rosiglitazone use was limited due to concerns about weight gain and fluid retention especially among comorbid cardiovascular patients [153].

In summary, metformin is promising at improving liver function, lipid metabolism, and insulin sensitivity in MASLD patients, particularly those with T2D. Its positive effects are independent of glucose control and work primarily through activated protein kinase (AMPK) activation, reduced lipogenesis, increased fatty acid oxidation, and anti-inflammatory mechanisms. However, clinical results remain mixed. Some studies report improvements in liver enzymes and fat accumulation, while others show minimal or even negative effects. Compared to newer antidiabetic agents such as TZD, GLP-1 receptor agonists, and SGLT2 inhibitors, metformin alone has a weaker effect on liver histology. Pioglitazone showed significant improvements in MASH histology, whereas rosiglitazone showed mixed results and is limited due to concerns about cardiovascular risks. Given the variability of clinical responses, further large-scale studies are needed to determine the optimal role of metformin in MASLD treatment, particularly when combined with other metabolic therapies.

#### 5.2.2. GLP-1 Receptor Agonists

GLP-1, a hormone produced by the L cells of the small intestine and proximal large intestine, plays a critical role in regulating plasma glucose levels. It increases insulin secretion in a glucose-dependent manner and inhibits glucagon secretion, which helps lower blood sugar [154]. In addition, GLP-1 has positive effects on metabolism, including improving IR, promoting weight loss through delayed gastric emptying, and reducing appetite [155].

However, GLP-1 is rapidly degraded by the enzyme DPP-4, which is abundantly expressed in the liver [156]. Elevated circulating DPP-4 levels are associated with more severe liver disease, particularly in individuals with MASLD [157]. This has led to the hypothesis that inhibiting DPP-4 could potentially improve the histological characteristics of MASLD and MASH (non-alcoholic steatohepatitis) by preserving GLP-1’s beneficial actions [158]. The positive effects of GLP-1 on MASLD can be both indirect and direct. Indirect effects occur by reducing appetite and delaying gastric emptying, GLP-1 may help to reduce body weight and insulin sensitivity, which in this case alleviates metabolic disorders commonly associated with MASLD. In addition, it acts directly as GLP-1 can act directly on hepatocytes (liver cells) to reduce triglyceride accumulation, improving liver fat levels and reducing inflammation [159].

One promising therapeutic approach involves tirzepatide, an agonist targeting both the GLP-1 receptor and the glucose-dependent insulinotropic polypeptide (GIP) receptor [160]. In a randomized phase 2 trial of patients with T2D, tirzepatide showed a significant reduction in MASH-related biomarkers including AST/ALT and procollagen III (a marker for liver fibrosis) [161]. In addition, tirzepatide increased levels of adiponectin, a hormone with antifibrotic and anti-steatogenic properties, which helps reduce liver fat and fibrosis. These results suggest that tirzepatide may have a dual benefit in treating MASLD and MASH, as it both improves metabolic factors such as IR and directly affects the liver.

In addition, liraglutide, a long-acting GLP-1 analog, has been approved by the FDA for the treatment of type 2 diabetes and has shown promise for treating MASH. In the Liraglutide efficacy and action in NASH (LEAN) study, liraglutide resolved MASH in 39% of patients compared with 9% in the placebo group (*p* = 0.02), with only minimal side effects, primarily gastrointestinal (e.g., diarrhea). Importantly, liraglutide was also associated with weight loss; nonetheless, it is not clear whether this improves liver histology or liraglutide itself had an independent effect on MASH. The study also suggested that liraglutide could reduce the fibrosis stage in patients [162].

#### 5.2.3. SGLT2 Inhibitors

Normally, the renal filtration threshold for glucose equals to 180 mg/dL, meanwhile when plasma glucose exceeds that number, glucose is eliminated in the urine [163].

SGLT consists of two transporters: SGLT1, which is present in both the intestines and kidneys, particularly in the thick part of the proximal tubule, and SGLT2, which is instead exclusively present in the renal tubule. Approximately 97% of glucose is absorbed upstream of the proximal tubule by SGLT2, while the remaining glucose is absorbed downstream by SGLT1 [164].

In hyperglycemic conditions, the kidneys increase their renal absorption capacity to a maximum of 600 g/day to prevent renal glucose loss. This effect, which is mediated by the SGLT cotransporter, not only results in an increase in the reabsorption of glucose but also of sodium and fluids, so that inhibiting this transporter would not only have a positive effect on blood sugar control but also on sodium homeostasis and water retention [165,166].

SGLT2 inhibitors have been developed to treat T2D. They inhibit the reabsorption of glucose in the kidneys and lower blood sugar levels [167]. For these pathophysiological reasons, SGLT2 inhibitors are promising and important therapeutic agents for patients with MASLD [168,169].

Numerous studies have shown that SGLT2 inhibitors reduce the risk of cardiovascular and kidney diseases, reduce body weight, and improve ALT plasma levels [24,170]. In addition, by reducing fat mass, SGLT2 inhibitors prevent the adipocyte release of inflammatory cytokines, thus reducing the inflammatory effects, which are a major cause of MASH progression [171].

Key studies include the e-LIFT study (2018), whereby Empagliflozin (10 mg) significantly reduced liver fat by 16.2% compared to 11.3% in the placebo group, thus improving liver enzymes (ALT) [172]; the study carried out by Kahl et al., which reported that Empagliflozin (25 mg) reduced liver fat content in T2D patients with MASLD, confirmed by magnetic resonance spectroscopy [173]; Shimizu et al.’s study, which reported dapagliflozin (5 mg/day) to result in significant reductions in liver fat and liver stiffness, which was investigated by transient elastography [174]. In addition, the study carried out by Ito et al. reported that ipragliflozin (50 mg) also improves liver steatosis, reduced liver enzymes, and lowers body weight and visceral fat [144], and finally, according to another study, over a period of six to twelve months, canagliflozin (100 mg/day) improved liver fat and histology, decreased IR, and decreased liver enzymes [175].

These studies show that SGLT2 inhibitors not only help control blood sugar but also improve liver function, reduce fat mass, and lower liver inflammation, which contributes to a reduced risk of MASH progression and other complications associated with T2DM and MASLD. This makes this drug class one of the most promising future therapies for the specific indication of MASLD.

#### 5.2.4. Statins

Statins are commonly used to treat lipid disorders, particularly those associated with elevated cholesterol levels, and are essential for treating CVD. They act as 3-hydroxymethyl-3-methylglutaryl-coenzyme A lyase (HMG-CoA) reductase inhibitors, reduce cholesterol synthesis, and lower serum cholesterol levels, which significantly reduces the morbidity and mortality of cardiovascular diseases [176]. Statins increase lipoprotein clearance primarily by reducing low-density lipoprotein (LDL) cholesterol and increasing LDL receptor expression [177]. They also have wider metabolic effects, including lowering TG, moderating post-meal TG increases, and increasing HDL cholesterol [178,179,180].

Statins not only lower cholesterol levels but also have pleiotropic effects. Lower levels of highly sensitive C-reactive protein (CRP) suggest that vascular inflammation has declined, and an increase in nitric oxide (NO) availability improves endothelial function [181]. Statins not only lower pro-inflammatory cytokines and increase platelet reactivity, but also support the healing process after myocardial ischemia—a reperfusion injury [182,183,184]. Overall, statins offer broad cardiovascular benefits, making them a cornerstone of contemporary cardiovascular treatment.

Statins are beneficial for patients with MASLD, especially those with concurrent CVD [185,186]. MASLD and CVD share common risk factors, and although liver-related mortality in MASLD is primarily due to complications such as cirrhosis and liver cancer, patients with MASLD are at higher risk of cardiovascular morbidity and mortality [187]. By blocking HMG-CoA reductase, statins lower cholesterol biosynthesis and effectively treat dyslipidemia, which is commonly seen in patients with non-alcoholic fatty liver disease. In particular, atorvastatin was found to improve liver enzyme levels and MASLD ultrasound markers while drastically reducing cardiovascular disease morbidity and mortality [185]. Statin therapy is particularly beneficial for statin-naïve MASLD patients with advanced fibrosis and increased cardiovascular risk [188,189,190]. Overall, statins are efficient in treating both MASLD and its associated cardiovascular risks, potentially saving lives as the disease progresses.

As for the safety part, statins are generally safe for treating MASLD and MASH despite concerns over potential liver toxicity [185,191,192]. Although statins can have side effects like myopathy, kidney, and liver dysfunction [193], these are rare, and recent studies suggest statins do not significantly elevate liver enzymes in MASLD patients. Large studies, including those focused on elderly and cirrhotic patients, indicate that statins are safe even in high-risk groups, reducing the risk of liver complications like HCC [194,195,196]. The National Lipid Association and recent guidelines advocate for statin use in MASLD/MASH patients with hypercholesterolemia, given their low hepatotoxic risk [197,198].

In addition, statins have shown promising effects in reducing inflammation and improving liver conditions in patients with MASLD and MASH. They exhibit anti-inflammatory properties by inhibiting pathways that lead to the activation of pro-inflammatory cytokines, such as IL-1β, IL-18, and TNF-α, which are linked to MASLD progression [199,200]. For example, atorvastatin reduces inflammation by blocking the NLRP3 inflammasome pathway, and long-term use can alleviate liver inflammation, steatosis, and hepatocellular damage [201].

Statins also have potential antifibrotic effects, which are critical to prevent liver disease progression. Studies suggest that statins, including simvastatin and fluvastatin, may reduce liver fibrosis by modulating enzymes such as nitric oxide synthase and inhibiting HSC activation [202]. Clinical evidence supports the assumption that taking statins correlates with a lower likelihood of advanced liver fibrosis and may reduce the risk of HCC in patients with MASH and advanced fibrosis [195,203]. Additionally, these drugs can help lower cholesterol levels and inflammation state by preventing cholesterol buildup in liver cells, reducing inflammation by inhibiting the activation of KCs and HSCs, both key players in fibrosis development. They also restore the health of liver sinusoidal endothelial cells (LSECs), reducing portal pressure and improving liver prognosis. Through Human serum paraoxonase-1 (PON1); statins restore serum PON1, an antioxidant enzyme, which reduces oxidative stress and helps reverse MASLD progression [189,204,205]. MASH is primarily caused by lipid peroxidation (LPO), which is reduced by increased PON1 activity. In addition, statins have anti-fibrotic and anti-inflammatory effects by blocking isoprenylation of small Guanosine-5′-triphosphate (GTPases) (such as Ras homolog family member A (RhoA) and renin–angiotensin system Ras) and reducing inflammation and fibrosis via signaling pathways such as Ras/extracellular signal-regulated kinase (ERK)1/2 and RhoA/RhoA kinase [206]. Statins also alter PPARs, particularly PPARα, which improves fatty acid metabolism and decreases fibrosis, IR, and liver steatosis. It improves liver health by inhibiting lipid synthesis and activating AMPK, a key metabolic regulator that helps reduce fat accumulation in the liver [207,208]. More research is needed to determine whether these drugs can help treat liver fibrosis by inducing ferroptosis, a type of iron-dependent cell death. According to recent research, changing the gut microbiota to a healthier profile may also help treat MASH and MASLD associated with obesity by reducing systemic inflammation [209].

In conclusion, and despite the growing evidence of the beneficial effects of statins on MASH and cardiovascular health, more clinical trials are needed to fully understand their role, particularly in relation to microbiota.

## 6. Emerging Therapies

### 6.1. FXR Agonists

Maintaining BA metabolic homeostasis is critical for preventing steatosis and liver damage, which may result from excessive cholesterol accumulation in the liver due to impaired BA conversion or cholestasis. In MASLD, BA metabolism is disrupted, resulting in elevated BA levels and an abnormal profile [210]. FXR plays a key role in regulating BA metabolism by controlling BA synthesis, excretion, and reabsorption, and significantly influencing MASLD pathogenesis. FXR is a nuclear receptor that is primarily activated by bile acids. Activating FXR suppresses the new synthesis of BAs from cholesterol, limits the circulating BA pool, and promotes bile acid transport from hepatocytes [211,212]. This receptor activation has several beneficial effects: improved insulin sensitivity, reduced hepatic gluconeogenesis, and increased expression of hepatic scavenger receptors, which accelerates reverse cholesterol transport [213,214,215]. Active FXR also causes small heterodimer partner (SHP) expression in the liver. Cholesterol 7 alpha-hydroxylase (CYP7A1) and cytochrome P450, family 8, and subfamily B, polypeptide 1 (CYP8B1), enzymes involved in BA synthesis, are then suppressed by SHP through its interaction with liver receptor homolog 1 (LRH-1) [216]. Additionally, activated FXR inhibits Multitasking Na+/Taurocholate Cotransporting Polypeptide (NTCP) and Organic anion transporting polypeptides (OATP1) to stop BA reuptake and increases the expression of bile salt export pump (BSEP) to facilitate BA secretion into bile ducts [217,218,219]. To further aid in the excretion of BAs and liver protection, FXR also activates enzymes involved in BA modification, including Cytochrome P450 3A4 (CYP3A4) and Uridine diphosphate (UDP)-glucuronosyltransferase [220]. Through the induction of fibroblast growth factor 15/19 (FGF15/19), intestinal FXR activation also aids in the restoration of BA homeostasis in MASLD and MASH [221,222]. The FGFR4 is bound by this protein, which decreases BA secretion by the liver. This activation of the ERK and Jun N-terminal kinase (JNK) signaling pathways prevents the synthesis of CYP7A1 and BA [223,224]. In addition, FXR activation helps reduce liver lipids by lowering the levels of mono- and polyunsaturated fatty acids (MUFA and PUFA) [214]. FXR also stimulates the expression of silent information regulator sirtuin 1 (SIRT1), which inhibits fatty acid synthesis and promotes oxidation; loss of SIRT1 results in MASLD. FXR and SIRT1 interact to regulate each other, with SIRT1 increasing FXR stability [225].

Since FXR activation has shown a protective effect in the treatment of MASLD, efforts are being made to develop various FXR agonists. However, most of these agonists, such as obeticholic acid (OCA), vonafexor, and tropifexor, cause side effects such as itching and increased LDL cholesterol levels that hinder their approval and use [226,227]. Research indicates that pruritus is linked to increased serum IL-31 levels caused by FXR activation in the liver [228].

OCA, an FXR agonist derived from chenodeoxycholic acid, has shown promise in the treatment of non-alcoholic steatohepatitis (MASH) and has improved the histological characteristics of MASH, particularly with regard to the NAFLD Activity Score (NAS) [229,230]. In phase 2a studies, OCA resulted in weight loss, increased insulin sensitivity, and reduced liver inflammation and fibrosis in patients with diabetes and MASLD. The phase 2b Farnesoid X Receptor Ligand Obeticholic Acid in Nonalcoholic Steatohepatitis Treatment (FLINT) trial involving 283 MASH patients showed that 45% of patients treated with 25 mg OCA daily had a significant reduction in NAS without the fibrosis worsening, compared with 21% in the placebo group. OCA also reduced fibrosis (35% versus 19%), steatosis, lobular inflammation, and hepatocyte balloon formation, leading to improvements in body weight and ALT levels [231]. OCA is currently being investigated in a phase 3 trial (REGENERATE) for its effectiveness in MASH for fibrosis and compensated cirrhosis due to MASH [232]. Nonetheless, one of the phase 2 studies was interrupted as the primary endpoints were not met by 18 months [233].

Alternative strategies include intestinal FXR-targeting FXR antagonists like Tauro-α-muricholic acid (T-α-MCA) and Gly-MCA, which have been shown to improve MASLD. Other substances that selectively block intestinal FXR and provide a better therapeutic profile than current treatments, like F6 and V023-9340, are presently under development [234,235]. In addition, drugs and natural extracts that indirectly target FXR are being researched. Plant compounds such as emodin and caffeic acid phenethyl ester (CAPE) activate FXR signaling pathways and have been shown to be effective in improving liver steatosis and MASLD [236,237]. Reused drugs such as disulfiram (DSF) and hydrogen sulphide (H2S) are also being investigated for their potential to increase FXR activity and improve lipid metabolism in MASLD/MASH patients [238,239]. Probiotics such as Akkermansia muciniphila and Bifidobacterium bifidum are also promising to regulate FXR and alleviate MASLD [240].

### 6.2. Acetyl-CoA Carboxylase (ACC) Inhibitors

De novo lipogenesis (DNL) plays a critical role in the development of MASH, in which excessive triglyceride accumulation leads to steatosis, lipid toxicity, inflammation, and fibrosis [241]. ACC inhibitors, which target DNL and promote mitochondrial fatty acid oxidation, are being explored as potential treatments for MASH [242]. These carboxylases have two isoforms, ACC1 and ACC2, which are different. The ACC2 of mitochondrial membrane controls fatty acid oxidation, whereas ACC1 primarily promotes fatty acid synthesis in the cytosol [243].

In healthy lean individuals, DNL contributes approximately 10% to liver lipids, but in obese individuals, this figure rises to 10–20%, and in patients with MASLD, to 25–40% [244]. DNL is primarily regulated at the transcriptional level, with insulin activating the SREBP1c transcription factor, which upregulates genes involved in fatty acid synthesis. Similarly, glucose uptake promotes carbohydrate-responsive element-binding protein (ChREBP), another transcription factor that enhances the biosynthetic gene expression of FA. Studies show that important DNL genes such as those for SREBP1c and DNL enzymes are upregulated in MASLD patients [245].

While there are limited data on the regulation of ACC isoforms in MASLD, studies indicate that ACC1 expression is increased in these patients [246]. MASLD may be exacerbated by high-glucose diets and sugar-sweetened beverages (SSBs), particularly those that contain fructose, which are associated with elevated DNL. Fatty acid synthase and acetyl-CoA carboxylase alpha (ACACA) (which codes for ACC1) are two genes in the DNL pathway that have been demonstrated to be upregulated by fructose consumption [247]. This suggests that the rise in fructose consumption, particularly through SSBs, may be a significant factor in the growing prevalence of MASLD in western countries [248].

Several ACC inhibitors, including Pfizer’s PF-05175157 [249], (MSD’s MK-4074 [250], Gilead’s GS-0976 (Firsocostat) [246,251], and Pfizer’s PF-05221304 [243,252], have shown promising results in reducing hepatic DNL, but also raised concerns about side effects, particularly hypertriglyceridemia (elevated TGs). For example, PF-05175157 was discontinued due to a reduction in platelet counts [253], while GS-0976 and PF-05221304 showed improvements in liver fat but also led to increased serum triglycerides in some patients [249,254]. This unexpected side effect may result from a deficiency of PUFAs in the liver, triggering a cascade of metabolic changes [246]. To address these issues, combination therapies are being tested. For instance, combining PF-05221304 with PF-06865571 (a diacylglycerol acyltransferase (DGAT) 2 inhibitor) or GS-0976 with Fenofibrate has shown potential in reducing hypertriglyceridemia and improving liver health in MASLD patients [254,255].

### 6.3. Thyroid Hormone Receptor (THR)-β Agonists

Thyroid hormones, which are generated by the thyroid and hypothalamus, control lipid metabolism. Thyroxin 3 (T3) binds to THR-α in the heart and bones and THR-β in the liver to affect gene transcription [256]. Studies show that hypothyroidism and elevated thyroid-stimulating hormone (TSH) levels are associated with an increased risk of MASLD [257]. Thyroid hormones affect liver function by promoting β-oxidation of fatty acids and reducing lipid accumulation in the liver through mechanisms such as autophagy, lipophagy, and mitochondrial biogenesis [258,259].

Studies on MASH in mice suggest that intrahepatic hypothyroidism plays a role in the development of the disease, with low thyroid hormone levels associated with increased inflammation and liver fat [260,261]. The conversion of T4 into the active hormone T3 is facilitated by the liver enzyme (Type 1 Iodothyronine Deiodinase) DIO1, but with advanced MASH, its activity decreases, which further impairs lipid metabolism. Since THR-β controls important metabolic pathways and is highly expressed in hepatocytes, it is a crucial target for the development of therapies for lipid-associated liver diseases such as MASLD and MASH [262].

Resmetirom is an experimental THR-β agonist that is highly selective for THR-β over THR-α and is 28 times more selective [262]. This agonist uses the OATP1B1 receptor for hepatocyte uptake and strongly activates THR-β [263,264]. It also supports DIO1’s function in converting T4 to T3 by upregulating it. This mechanism is further supported by the low serum T4 levels in patients treated with resmetirom [265]. Due to its specificity and effects on lipophagy, mitophagy, mitogenesis, and β-oxidation in hepatocytes, several studies have been conducted with the drug. The results were promising and reduced LDL-C, TGs, and apoloprotein B without affecting TSH and free T3 levels. In addition, another phase 2 study with this agonist showed a significant reduction in liver fat compared to a placebo and an improvement in liver enzyme levels, lipid profiles, and inflammatory markers. Thus, resmetirom was approved by the US FDA on 14 March 2024 for the treatment of MASH. Safety profiles were favorable, there were no signs of adverse effects from THR-α activity, and thyroid function remained stable [266,267,268].

### 6.4. Antibiotics

The gut–liver axis plays a significant role in the mechanism of MASLD. Dysbiosis as discussed above has been associated with fat accumulation, liver inflammation, and fibrosis. Since it has the potential effect on gut microbiome, the use of antibiotics has gained a lot of attention in recent years. It may help restore microbial balance by modifying gut microbiome. Recent studies suggest specific gut-targeting antibiotics could reduce harmful substance translocation from the gut to liver. One of the investigated antibiotics is Rifaximin, a derivative of Rifamycin but gut-selective antibiotic with minor systemic effect. It has FDA approval for treating irritable bowel syndrome (IBS) [269,270,271,272], hepatic encephalopathy for reducing recurrent episodes [273,274,275,276], and traveler’s diarrhea [277,278,279,280]. A recent study suggested that rifaximin affects liver inflammation and intestinal microbiome, and reported efficacy and safety of the drug through a reduction of serum endotoxin, cytokine18 CK18, proinflammatory cytokine (PROC 3,4,8), and fat scoring of MASLD [281]. We advise the investigation of this drug since it is gut selective with almost no systemic bioavailability (less than 0.04% in blood stream) and could avoid possible antibiotics resistance in the long-term effects [278,282,283]. Nonetheless, the therapeutic use of antibiotics in MASLD remains controversial, with some experts cautioning against their use due to concerns over long-term effects on the microbiome.

In conclusion, while antibiotics, especially those like rifaximin, show promise as a therapeutic tool in modulating the gut–liver axis in MASLD, further research is needed to better understand their efficacy and safety in this context. The development of targeted microbiome-based therapies integrated with multi-omics profiling harnessing the benefits of antibiotics without disrupting the microbiome balance may provide a new avenue for MASLD treatment in the future.

### 6.5. Novel Anti-Inflammatory and Anti-Fibrotic Agents

Since inflammation and fibrosis are key components in the progression from MASLD to MASH and cirrhosis, new agents are currently being investigated that target these processes. These agents target specific inflammatory pathways (e.g., C-C chemokine receptor type (CCR) (CCR2/CCR5 antagonists) or fibrosis-promoting processes (e.g., galectin 3 inhibitors, anti-Lysyl oxidase Homolog (LOXL2 antibodies). These agents reduce inflammation and fibrosis and aim to stop or reverse the progression of MASLD. Although reducing systemic inflammation is primarily aimed at liver diseases, it could also have a positive effect on METS components [284].

In addition to the various current therapeutic targets, one approach includes FGF21. It has been reported to have anti-fibrotic effects and improve metabolic status. In the BALANCED study, the use of efruxifermine effectively and safely reduced liver fat in patients with MASH cirrhosis [285]. In addition, FGF-21 (efruxifermine) replaced MASH in the HARMONY study, a double-blind randomized controlled phase 2 trial, and improved liver fibrosis in patients with F2 or F3 fibrosis [286]. In addition, in another phase 2 trial, FGF-21 agonist in combination with GLP-1 significantly improved noninvasive markers of MASH-related disease [287]. To add more, A current ongoing clinical trial in a phase 2 phase is investigating the role of FGF19 analogue as a reversal drug in MASH and liver fibrosis; further trials are currently underway [288].

In summary, the FGF-21 agonist, the FGF-19 analog, and the CCR2/CCR5 antagonist have great therapeutic potential for the treatment of MASH and MASLD. However, thorough phase 3 clinical trials are essential to determine their place in clinical practice and to develop long-term safety and efficacy profiles in a larger patient population.

## 7. Future Directions

### 7.1. Precision Medicine Approaches

A one-size-fits-all strategy might not be the best option due to the heterogeneity of MetS and MASLD. The onset and progression of MetS and MASLD are influenced by both genetic and environmental factors. The goal of precision medicine techniques is to customize care according to each patient’s unique genetic profile, environmental exposures, and disease characteristics [289].

Omics approaches are recent evolving approaches in the field of clinical research. The integration of multi-omics modalities could provide more thorough understanding of the underlying mechanisms of liver disease, especially MASH. Multi-omics involving research recently investigated MASH and alcoholic liver disease and cirrhosis. The use of genome, epigenome, transcriptome proteome metabolome, and microbiome profiling in combination with a clinical scoring system can shed light on complex relationships and individual therapeutic potentials. A multi-omics study of MASLD published in nature genetics revealed 18 sequence variants associated with MASLD, 4 variants associated with cirrhosis, and surprisingly two protective variants. Nonetheless, the phenotype was not similar between these shared variants which suggests more than one biochemical pathway that leads to MASLD [290]. Another notable study, featured in *Frontiers in Microbiology*, employed multi-omics techniques to investigate the biological mechanisms of MASLD. By analyzing genomic, transcriptomic, and metabolomic data, the researchers identified specific microbial signatures and metabolic alterations associated with the disease [291].

These advancements in multi-omics approaches have significantly enhanced our understanding of non-alcoholic fatty liver disease (MASLD), hopefully facilitating the identification of novel biomarkers and therapeutic targets. Furthermore, using multi-omics techniques in combination with already-proven therapy in context with the resolution of fibrosis or normalization of hepatocellular enzyme, which have not been done before, may at last discover the needed metabolite or protein than can be used as a primary or secondary preventive screening tool. Thus, we encourage revolutionary approaches in the field of hepatology research by combining different types of studies.

### 7.2. Microbiome Modulation

Novel therapeutic approaches that target the microbiome have surfaced as promising tools for managing MASLD and enhancing liver function as its prevalence rises worldwide.

Probiotic use is among the most researched methods for modifying the gut microbiota. These are live microorganisms that provide the host with health benefits when given in sufficient amounts [292,293]. In order to stop harmful endotoxins and bacterial products from leaking into the bloodstream, probiotics improve the intestinal barrier [294], encourage the growth of good bacteria [295], and restore the balance of the gut microbiota [296]. In individuals with MASLD, probiotics can help reduce inflammation, improve insulin sensitivity, and reduce liver fat accumulation. Several studies have demonstrated that specific strains of probiotics, such as Lactobacillus and Bifidobacterium, can modulate metabolic parameters and improve liver enzymes in patients with MASLD [297,298]. For instance, a randomized clinical trial reported that probiotic supplementation improved liver function and reduced inflammation in patients with MASLD [299].

In addition to probiotics, prebiotics—non-digestible food components that selectively promote the growth of beneficial bacteria—have been investigated as a therapeutic intervention for MASLD [300]. Prebiotics, particularly dietary fibers and oligosaccharides, influence the composition of gut microbiota by stimulating the growth of beneficial bacteria such as Bifidobacteria and Lactobacilli This in turn leads to the production of short-chain fatty acids (SCFAs) such as butyrate, acetate, and propionate, which have been shown to have anti-inflammatory and anti-fibrotic effects [301,302]. Another systematic review emphasized the beneficial effects of prebiotics in regulating liver fat accumulation and improving liver function [303]. Furthermore, a high-fiber diet enriched with prebiotics led to a reduction in liver fat content and improved insulin sensitivity in mice with MASLD [304].

Synbiotics, which combine both probiotics and prebiotics, are another therapeutic approach that has gained traction. The rationale behind using synbiotics is that they can provide both live beneficial bacteria and substrates that enhance their growth and activity, thereby offering a more comprehensive approach to restoring gut microbiome balance [305,306]. Furthermore synbiotic supplementation improved metabolic parameters, liver enzymes, and inflammatory markers in MASLD patients [307]. The combined action of probiotics and prebiotics in synbiotics may offer synergistic effects in restoring gut microbiota homeostasis, reducing hepatic fat content, and preventing liver damage [308].

Fecal microbiota transplantation (FMT) has emerged as a promising therapeutic approach for MASLD. This treatment involves introducing donor stool into the patient’s gut with minimal manipulation to restore the patient’s disrupted gut microbiome [309]. The onset and progression of MASLD and MASH have been linked to dysbiosis, or changes in the composition of the gut microbiota. Increased intestinal permeability brought on by dysbiosis may enable toxic substances to enter the liver through the portal circulation, causing inflammation and liver damage. By restoring a balanced gut microbiota, FMT may lessen intestinal permeability and lessen inflammation in the liver [310].

Recent studies have explored its potential to restore gut microbiota balance and improve liver function. A randomized controlled trial by Craven et al. (2020) investigated the effects of allogenic FMT in MASLD patients. The study found that while FMT did not significantly alter IR or hepatic fat content, it notably reduced small intestinal permeability, suggesting a potential benefit in restoring gut barrier integrity [311].

Furthermore, research by Xue et al. (2017) demonstrated that FMT could decrease hepatic fat accumulation by improving gut microbiota dysbiosis, thereby attenuating fatty liver disease. The study highlighted significant differences in clinical features and gut microbiota between lean and obese MASLD patients, with FMT showing more pronounced effects in lean individuals [296]. Additionally, a comprehensive review by Del Barrio et al. (2023) emphasized the therapeutic potential of FMT in MASLD. The authors noted that FMT could correct intestinal bacterial imbalances, thereby preventing the development and progression of MASLD [312]. FMT represents a novel approach to managing MASLD and MASH by targeting gut microbiota dysbiosis. Although preliminary studies are promising, comprehensive clinical trials are necessary to fully establish their role in the treatment of these liver diseases.

### 7.3. The Role of Artificial Intelligence in Prediction, Diagnosis, and Treatment

Artificial intelligence (AI) techniques are enhancing disease prediction for MASLD, and MetS. Machine learning models trained on clinical and laboratory features can identify patients at risk with high accuracy [313,314,315]. For example, a machine learning (ML)-based model using routine lab markers achieved an area under the receiver operating characteristic curve (AUROC) ~0.81–0.84 in validation for MASLD detection [314,316]. In MetS, AI algorithms (e.g., gradient boosting, support vector machine (SVM)) have been applied to large datasets to predict syndrome onset, enabling early lifestyle interventions [317]. These predictive models can flag high-risk individuals before clinical diagnosis, supporting preventive care planning.

Deep learning applied to medical imaging has improved MASLD diagnosis. AI-assisted ultrasound, for instance, can noninvasively detect hepatic steatosis with sensitivity and specificity ~0.97 and 0.98, markedly outperforming conventional clinical scoring methods [318]. Advanced models analyze ultrasound, computed topography (CT), or magnetic resonance imaging (MRI) scans to quantify liver fat and fibrosis, reducing the need for biopsy. In one study, a deep-learning index derived from ultrasound images distinguished moderate-to-severe MASLD with AUROC ≈ 0.966. Such image-based AI tools can stratify disease severity and even predict complications (like cirrhosis or HCC risk) from routine scans, aiding radiologists in faster, more accurate MASLD diagnosis.

AI is transforming histological assessment of liver biopsies in MASLD. Digital pathology algorithms can evaluate biopsy slides for steatosis, inflammation, ballooning, and fibrosis with high concordance to expert pathologists. In a recent study, a deep neural network trained on hundreds of NASH biopsy images matched pathologist scores for key features and fibrosis stage. Similarly, machine learning classifiers have achieved ~89% accuracy in identifying histologic fat and architectural features on whole-slide images [319]. These tools reduce inter-observer variability and offer rapid, standardized MASH activity and fibrosis scoring. AI-based digital pathology is already being used in clinical trials to quantify treatment response (e.g., fibrosis regression) more sensitively than traditional pathology [319]. Overall, automated histopathology analysis promises to streamline MASLD diagnosis and staging in both research and practice.

AI-driven approaches are accelerating drug discovery for MASLD/MetS. By mining big data and using predictive modeling, AI can identify novel therapeutic targets and drug candidates. For example, an AI-guided ensemble learning strategy was used to discover new nonsteroidal FXR agonists for MASH treatment [319,320]. An integrated machine learning which poses filters with virtual screening to overcome protein-flexibility issues, has successfully uncovered a potent FXR agonist that traditional methods missed [321]. AI is also being applied to optimize clinical trial design and personalize therapy. In MetS, AI models can predict which patients will respond to interventions (diet, exercise, or medications), supporting more tailored treatment plans [317]. Additionally, “digital twin” simulations and AI-powered organ-on-a-chip systems are emerging to test drug effects in silico. Early studies show AI-enhanced liver micro-physiological models can detect drug-induced liver changes and toxicity faster and more reliably [322,323,324]. Such innovations accelerate the development of effective pharmacological therapies for MASLD, a field that until recently had limited approved drugs.

AI integration in clinical workflows supports personalized medicine for patients with MASLD and MetS. By analyzing electronic health records (EHR), lab results, imaging, and even genomic data, AI can provide clinicians with decision support tailored to the individual. For instance, algorithms combining EHR variables with imaging data have been shown to accurately predict liver fibrosis and MASH, potentially obviating some biopsies [315,325,326,327]. AI-based clinical decision support systems can risk-stratify MASLD patients (e.g., identifying those likely to progress to cirrhosis) and suggest optimal management plans. In metabolic syndrome, ML models have been used to scan large healthcare databases to identify patients with undiagnosed MASH or diabetes risk, alerting providers to intervene earlier. One such study in a Veterans cohort which flagged 12% of at-risk patients as likely undiagnosed MASH, enabling targeted outreach [328]. This kind of AI-driven decision support can improve outcomes by ensuring timely interventions (e.g., weight loss programs or pharmacotherapy) for those who need them most. As these systems evolve, they incorporate more data (wearables, lifestyle, multi-omics) to further personalize care, aligning with precision medicine initiatives in metabolic diseases.

In summary, the pathophysiology of MASH and MASLD has been better understood. These diseases are now a major health problem worldwide, and their rising incidence is linked to rising rates of metabolic syndrome. The understanding of disease mechanisms has improved thanks to developments in imaging techniques, the discovery of genetic markers, and the investigation of inflammatory and metabolic pathways. Yet, there are still significant gaps, particularly in the areas of early intervention, disease stratification, and noninvasive diagnosis. Invasive liver biopsies are still required to accurately differentiate between simple liver steatosis and more severe forms of the disease, such as MASH or advanced fibrosis, which cannot be reliably differentiated using the current biomarkers. There is also a significant unmet need in clinical treatment, as existing therapies are not precise enough to target specific disease mechanisms.

The variability of disease progression in MASLD and MASH research is one of the biggest obstacles and could be a crucial part of future treatment. Simple steatosis can remain stable in some people, but in others, it can quickly progress to more serious forms, such as cirrhosis and hepatocellular carcinoma. This variation in disease progression and our incomplete knowledge of the underlying molecular causes underscore the need for more individualized therapeutic strategies. This trajectory is heavily influenced by genetic, epigenetic, and environmental factors, but little is known about how these factors work together. In addition, the dynamics of the gut–liver axis and its role in aggravating diseases, particularly through mechanisms such as metabolic endotoxemia and microbiota dysbiosis, add further complexity that has not yet been thoroughly investigated.

A hitherto unprecedented opportunity to close these gaps has arisen with the development of multi-omics technologies. Understanding the molecular mechanisms underlying MASLD and MASH can be achieved by integrating data from the fields of genomics, transcriptomics, proteomics, metabolomics, and microbiomes. These technologies shed light on the complex networks of protein interactions, metabolic pathways, gene expression, and microbial influences that lead to liver dysfunction and disease progression. These cutting-edge techniques enable researchers to find new therapeutic targets and biomarkers, which can lead to early detection and more individualized, successful treatments. In addition, by identifying novel drug pathways, multi-omics techniques can help develop targeted therapies that target the underlying causes of liver damage and not just its symptoms.

In parallel with these technological developments, it is crucial to conduct cooperative, multidisciplinary research. Clinical, molecular and bioinformatics expertise must be integrated into future research to fully exploit the potential of multi-omics. These types of partnerships will be critical to gain a better understanding of the various mechanisms underlying MASLD and MASH, including their response to treatment. More effective and economical healthcare solutions could result from identifying patient subgroups who would benefit most from specific therapies through a combined clinical and omics-based approach. The transition from research to clinical practice can also be accelerated by integrating artificial intelligence and machine learning tools to improve predictive modeling of disease progression and treatment outcomes.

Finally, research into MASLD and MASH could result in significant progress in our knowledge of disease mechanisms and the development of efficient therapies. We will gain important new information about the pathophysiology and evolution of these disorders as we continue to explore their complex, multifactorial nature using multi-omics platforms. With these developments, the prospect of precision medicine and more specialized treatment approaches is becoming ever more realistic. Realizing this potential, however, requires consistent, collaborative work to fill current knowledge gaps and promote creative, integrated strategies to address the global burden of MASLD and MASH.

## Figures and Tables

**Figure 1 ijms-26-03448-f001:**
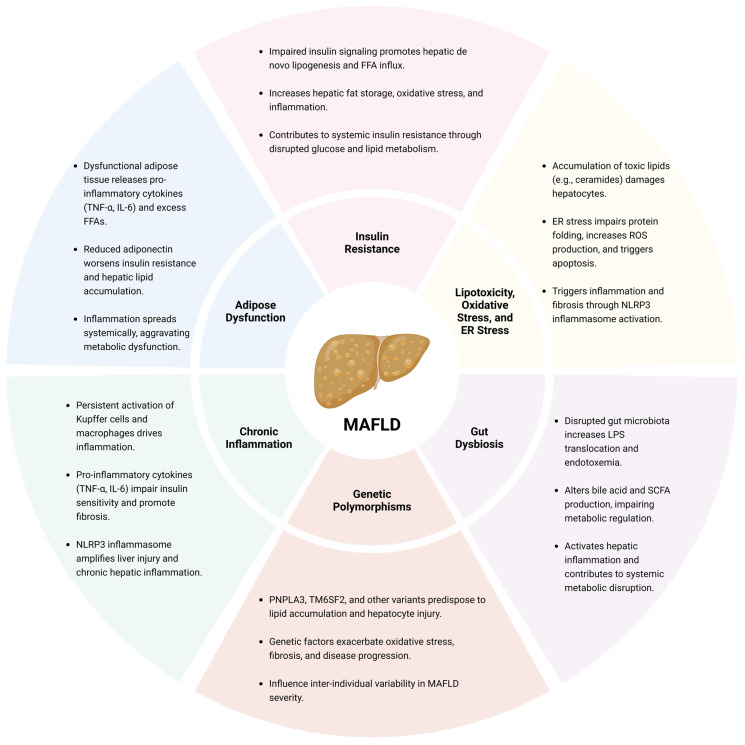
Mechanistic contributors to metabolic-associated fatty liver disease (MASLD). The figure highlights six interconnected mechanisms driving the pathogenesis of MASLD, including IR, adipose tissue dysfunction, lipo-toxicity and oxidative stress, gut dysbiosis, genetic polymorphisms, and chronic inflammation. These pathways collectively exacerbate hepatic fat accumulation, inflammation, and fibrosis, illustrating the multifactorial nature of MASLD and its systemic impacts. Understanding these mechanisms is crucial for developing targeted therapeutic strategies.

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
