# Peer review of "Hidden in the Fat: Unpacking the Metabolic Tango Between Metabolic Dysfunction-Associated Steatotic Liver Disease and Metabolic Syndrome"

_ijms, 2025, doi:10.3390/ijms26073448_

Round 1
Reviewer 1 Report
Comments and Suggestions for Authors
comments in the attached document

comments in the attached document
Author Response
Thank you for your comments , please open the attached file.
Thank you again

Reviewer 2 Report
Comments and Suggestions for Authors
The manuscript presents an interesting and well-structured review. It is well written and supported by 318 references, more than 50% of which have been published within the last five years. This ensures that the information is up to date and relevant.
The references are well selected and provide strong support for the manuscript's objectives.
General Comments:
- Please carefully review the use of "for" and "to" throughout the document.
- There are minor issues regarding the placement of periods (".") before and after in-text citations. For example, in lines 35, 37, 39, and 48. Please revise the entire document, as this issue appears repeatedly.
- Ensure the appropriate use of uppercase and lowercase letters following commas and periods, as some inconsistencies appear to be typographical errors.
The manuscript is particularly promising in its discussion of molecular mechanisms. However, the section on artificial intelligence only vaguely introduces the topic. It could be better developed and potentially incorporated into the Future Directions section, where its potential applications in the field could be further explored.
Overview of the Manuscript Sections:
The article provides a comprehensive review structured into the following sections:
- Introduction
- Change of Nomenclature – Addressing the transition from NAFLD/NASH to MASLD/MASH.
- Underlying Mechanisms (Figure 1)
- Insulin resistance (consider adding a summary to maintain consistency with other subsections).
- Adipose tissue dysfunction and proinflammatory state.
- ER stress and oxidative stress as amplifiers of liver injury.
- Gut microbiota dysbiosis and metabolic endotoxemia.
- Innate immune response and hepatic inflammation.
- Fibrosis pathways and liver disease progression.
- Clinical Significance
- Cardiovascular morbidity and mortality.
- Progression of liver disease.
- Type 2 diabetes mellitus.
- Extrahepatic complications, including:
- Chronic kidney disease.
- Obstructive sleep apnea.
- Osteoporosis.
- Implications for clinical management.
- Therapeutic Strategies and Future Directions
- Lifestyle modifications.
- Pharmacological interventions, including:
- Insulin sensitizers.
- GLP-1 receptor agonists.
- SGLT inhibitors.
- Statins.
- Emerging Therapies
- FXR agonists.
- Acetyl-CoA carboxylase inhibitors.
- THR-β agonists.
- Antibiotics.
- Novel anti-inflammatory and anti-fibrotic agents.
- Future Directions
- Precision medicine approaches.
- Microbiome modulation.
- Potential applications of artificial intelligence in disease prediction, diagnosis, and treatment optimization.
Although the document specifies the use of the updated terms MASLD and MASH, it continues to use the previous terminology (NAFLD and NASH). If the nomenclature has officially changed and is emphasized in the text, what is the rationale for retaining the outdated terminology?
Comments on the Quality of English LanguageNot able to assess, it seems well written. However, there some issues that need attention before publication associated with typographical mistakes.
Reviewer 3 Report
Comments and Suggestions for Authors
In this review the authors endeavor to describe the current knowledge about NAFLD, a condition which has multiple pathogenetic mechanisms, is still poorly understood and indeed deserves attention from the scientific community due its increasing prevalence worldwide.
The review is well-written, comprehensive, informative to experts in the filed but also to a general scientific readership.
I have just a few suggestions, which account for minor criticisms.
1-The title: though catchy, it conveys the impression of a hyper-acute progression of the disease, which is not the case. Perhaps something like “fat but not beautiful”, or “too much not always a good thing” would be less impressive, but not misleading regarding disease progress.
2-Include a paragraph on genetic predisposition (see 10.1038/s41588-023-01497-6), and a comment on the very significant prevalence of NAFLD in Latin America, which also impacts Western countries due to immigration (10.1097/HEP.0000000000000004).
3-Define abbreviations just once in the text, not in each and every paragraph.
4-Chapter 4 (Clinical Significance). Perhaps a better title could be found to describe the object of this paragraph, i.e. the impact of the MAFLD pathogenetic mechanisms on non-liver organs. “Extrahepatic co-morbidities” would be more descriptive. Also, in this chapter no mention is made of endocrine disorders associated with MAFLD; a hint at polycystic ovary syndrome and MAFLD should be added (10.3389/fmolb.2022.888194; 10.3390/biomedicines10112719; 10.1186/s12916-023-02775-0).
5-Please explain some unclear statements. Line 544 “Intermediary mechanisms between OSA and MAFLD seem to be similar”. This sentence is unclear, please explain. Lines 663-664; “increases insulin sensitivity of the peripheral” you mean extra-hepatic tissues? “Reduces production of basic liver glucose” what do you mean with “basic”? Line 678: “bound and free cholesterol” you mean HDL and LDL cholesterol? Line 732: “decreases and promotes ketogenesis”, which one of the two? Line 941: “secreted into the liver” what do you mean? Line 900: “PON1” acronym not described. Line 984: “these inhibitors”, they are carboxylases, not inhibitors.
6-Paragraph 5.2.1 contains many repetitions of the same concepts, it could be more concise.
7-Paragraph 5.2.3: the renal glucose filtration threshold is a glycemia of 180 mg/dL, not 180 g/day.
Comments on the Quality of English LanguageThere appears to be a different quality of the English in different paragraphs. I suggest a thorough re-reading by the authors, also to correct several spelling mistakes.
Round 2
Reviewer 1 Report
Comments and Suggestions for Authors
Please find some comments below regarding the revised version of your manuscript;
- It seems you haven't changed the NAFLD and NASH terms everywhere in the text. To reitarate, I suggest you revise this this to maintain consistency and change NAFLD to MASLD and NASH to MASH, even if they weren't reported as so in the original paper. You can refer to the change of the nomenclature earlier at your text in order to maintain a consistency. Additionally, you may cite the following reference in the methodology section: ''Hagström H, Vessby J, Ekstedt M, Shang Y. 99% of patients with NAFLD meet MASLD criteria and natural history is therefore identical. J Hepatol. 2024 Feb;80(2):e76-e77. doi: 10.1016/j.jhep.2023.08.026. Epub 2023 Sep 9. PMID: 37678723'', in the methodology section, which basically points out exactly this (that using MASLD instead of NAFLD is the same).
- Please review the text again for consistency in abbreviation usage. For example, "IR" is sometimes written as both "IR" and "insulin resistance." The correct approach is to introduce the full term followed by its abbreviation the first time it appears, such as "insulin resistance (IR)," and then use only the abbreviation (IR) throughout the rest of the text. This should be applied consistently to all abbreviations in the manuscript as it is the point of using them.
Author Response
Dear reviewer,
Thank you again for your constructive advice.
1) Comment 1, We changed the NAFLD to MASLD in the whole manuscript. previously the idea was to use NAFLD and after the section of change on nomenclature to use MASLD. We changed it according to your advice.
We also added the Letter to the editor by hangstrom regarding the 99.7% fit of nafld and masld criteria in the methods section.
2) Changed Insulin Resistance in the whole manuscript ( kept 2 first one with the abbreviation, and the title of the section in underlying mechnism.
thank you very much again